# SANTA: Source Anchoring Network and Target Alignment for Continual Test Time Adaptation

**Goirik Chakrabarty**                                        *goirik.chakrabarty@students.iiserpune.acin*
*Indian Institute of Science, Education and Research, Pune*

**Manogna Sreenivas**                                                    *manognas@iisc.ac.in*
*Indian Institute of Science, Bangalore*

**Soma Biswas**                                                          *somabiswas@iisc.ac.in*
*Indian Institute of Science, Bangalore*

**Reviewed on OpenReview:** *https://openreview.net/forum?id=V7guVYzvE4*

## Abstract

Adapting a trained model to perform satisfactorily on continually changing test environments is an important and challenging task. In this work, we propose a novel framework, SANTA, which aims to satisfy the following characteristics required for online adaptation: 1) can work effectively for different (even small) batch sizes; 2) should continue to work well on the source domain; 3) should have minimal tunable hyperparameters and storage requirements. Given a pre-trained network trained on source domain data, the proposed framework modifies the affine parameters of the batch normalization layers using source anchoring based self-distillation. This ensures that the model incorporates knowledge from the newly encountered domains, without catastrophically forgetting the previously seen domains. We also propose a source-prototype driven contrastive alignment to ensure natural grouping of the target samples, while maintaining the already learnt semantic information. Extensive evaluation on three benchmark datasets under challenging settings justify the effectiveness of SANTA for real-world applications. *Code here: https://github.com/goirik-chakrabarty/SANTA*

## 1 Introduction

Deep learning has achieved phenomenal success in several computer vision tasks like classification, object detection, segmentation, etc (Deng et al., 2009; Ren et al., 2015; He et al., 2017; Chen et al., 2018; Everingham et al., 2010). But it is well known that these models tend to perform poorly when the test data comes from a different distribution compared to the training data Ovadia et al. (2019). Unsupervised Domain Adaptation (UDA) techniques (Ajakan et al., 2014; Saito et al., 2018) have shown promising results in this scenario, but they require access to unlabeled target data along with the labelled source data, which may be difficult due to privacy concerns, storage constraints, etc. In addition to having a different distribution compared to the training data, the test data distribution may dynamically vary with time. For example, in autonomous driving, the model trained using data captured in clear weather can encounter cloudy weather followed by heavy rain, etc. during deployment. Thus, continually adapting the model during test-time is critical for the model to perform well in changing scenarios.

Test-time adaptation of trained models has thus emerged as an important research area, where an off-the-shelf trained model is adapted to the testing data as and when they are encountered. Majority of successful frameworks for this task, (Wang et al., 2021; Boudiaf et al., 2022), assume that the test data belongs to a single domain, which is a restrictive assumption for practical applications. Very recently, researchers have started to look at the continual test time adaptation setting (Wang et al., 2022a; Döbler et al., 2023; Brahma & Rai, 2023), where the target distribution can change over time.

In this work, we propose a novel framework, termed **S**ource **A**nchoring **N**etwork and **T**arget **A**lignment (SANTA), for the task of continual test-time domain adaptation. We believe that, for the model to be practically useful in an online setting, it should satisfy the following requirements: *1) For online adaptation, the models should work effectively with different (preferably small) batch sizes, while also being computationally efficient; 2) The updated model should continue to work well on the source domain; 3) The framework should require less storage and minimal tunable hyperparameters, since a validation set is usually not available during test-time.* With this motivation, we propose to use source anchoring based self-distillation, which ensures that the model robustly adapts to the incoming data, while not forgetting the source domain information. The proposed SANTA framework also utilizes contrastive learning to ensure better model generalizability to unseen domains. Further, we utilize the source prototypes for alignment of the target features to the corresponding source data, which helps preserve the semantic information learnt using the source. We propose to only update the Batch-Normalization (BN) affine parameters like TENT Wang et al. (2021), as it avoids overfitting on the small amount of target data and also reduces the storage requirements. This simple, yet effective framework helps us take a step forward in achieving all the objectives mentioned earlier. Extensive experiments on three large-scale benchmark datasets, namely CIFAR-10C, CIFAR-100C and ImageNet-C Hendrycks & Dietterich (2019) for different challenging and realistic scenarios justify the effectiveness of the proposed SANTA framework. To summarize, the contributions of this work are as follows:

- We propose a novel SANTA framework for the task of continual test-time domain adaptation.

- The proposed framework takes a step forward in overcoming some of the important challenges in a practical test-time adaptation setting.

- We show that the proposed source-based anchoring along with contrastive target alignment guided by source prototypes can be effectively utilized to robustly update the model under dynamically changing test conditions.

- Extensive evaluation on several benchmark datasets for challenging scenarios justify its effectiveness.

We now discuss the related work, followed by the proposed method and evaluation.

## 2 Related works

The proposed approach is inspired by several seminal works in different areas. Here, we provide pointers to some of the related work in literature.

**Source Free Domain Adaptation:** Source Free Domain Adaptation (SFDA) methods aim to adapt a model trained using source data to a new domain using only the unlabeled target data (Xia et al., 2021; Huang et al., 2021; Yang et al., 2021a; Liang et al., 2020; Kundu et al., 2020; Li et al., 2020; Tian et al., 2021; Yang et al., 2021b), Some of these methods, such as SHOT (Liang et al., 2020), use a pseudo-labelling strategy to maximize information and minimize entropy, while others, such as Li et al. (2020), use generative models to enhance model performance on the target domain by generating target-style images. GSFDA Yang et al. (2021b) aims to activate different channels within a network for different domains while also taking into account the local data structure. $A^2$-Net Xia et al. (2021) is another method that utilizes different classifiers to align the two domains using adversarial training. BUFR Eastwood et al. (2021) propose a Bottom-Up Feature Restoration scheme where they store a lightweight source feature distribution and then adapt the feature-extractor such that the target feature distribution realigns with that saved on the source.

**(Continual) Test time adaptation:** Test Time Adaptation (TTA) is an online variant of SFDA that adapts the model at test time using small batches of test data, as and when they become available. Here, the model's parameters or architecture are usually adjusted to handle the differences between the two domains better, thereby improving its performance on the target domain (Wang et al., 2021; Shen et al., 2022; Boudiaf et al., 2022; He et al., 2021; Schneider et al., 2020). Some of these methods focus on modifying the original architecture during the source training like TTT Sun et al. (2020), TTT++ Liu

et al. (2021) which trains the model on supervised and self-supervised tasks using source data. During testing, the self-supervised module is fine-tuned on the target data to improve performance. MT3 Bartler et al. (2022) employs meta learning and self supervision to effective adapt a model during test time. After that, methods such as EATA Niu et al. (2022) use some fraction of source data for identifying the effect of domain shift and update the model accordingly during test time. Recently, several researchers are focusing on the fully test time adaptation setting (Wang et al., 2021; Schneider et al., 2020; Zhang et al., 2022; Mummadi et al., 2021), which does not assume any access to source data or the source training process making it more practical. TENT Wang et al. (2021) adopts entropy minimization objective for training the BN layers, while BN Stats Adapt Schneider et al. (2020) adjusts the BN statistics during test time to align the target with the source domain.

A more realistic scenario is handled by the recently proposed continual test-time adaptation protocol Wang et al. (2022a), where the trained model should continually adapt to a dynamic environment, i.e. the test domain can change over time. CoTTA Wang et al. (2022a) utilizes weight-averaged and augmentation-averaged predictions to reduce error accumulation and also stochastically restores a small part of the neurons to the source pre-trained weights during each iteration to avoid catastrophic forgetting. This allows for long-term adaptation of all parameters in the network while preserving source knowledge. Recent methods (Döbler et al., 2023; Brahma & Rai, 2023) which have achieved better performance than CoTTA also use a teacher student framework to update the adapting model. Specifically, RMT Döbler et al. (2023) uses symmetric cross-entropy loss in a teacher-student framework. PETAL Brahma & Rai (2023) use a data-driven parameter restoration technique as opposed to stochastic restoration in CoTTA.

Recently, several methods (Gan et al., 2022; Gong et al., 2022; Niu et al., 2022) have been developed which can aid the dynamic adaptation to test-data. However, these methods need to be optimised along with the model during training with source. Hence, we do not compare with these methods as, in our protocol, we do not assume access to source data.

**Knowledge and Self-distillation:** Knowledge distillation is a technique used to transfer knowledge from a large, complex model ("teacher" model) to a smaller, simpler model ("student" model) (Hinton et al., 2015; Gou et al., 2021). This is done by training the student model to mimic the predictions of the teacher model, which has already learned useful representations, rather than training the student model on the original labeled data. CoTTA also utilizes distillation method to enhance the adaptation to new domains, which involves the implementation of a teacher model to make accurate predictions based on the student model.

A variant termed as self-distillation or self-knowledge distillation Yuan et al. (2019) or self learning Rusak et al. (2022), involves training a model to mimic its own predictions. The framework in Yang et al. (2019) shows that using a model from a previous epoch to train the same model in future epochs can increase the training efficiency and accuracy of the model.

**Contrastive learning:** Contrastive learning (Chen et al., 2020; Khosla et al., 2020; Caron et al., 2020) has shown tremendous improvement in learning visual features for various downstream tasks. Because of its robustness, researchers have used it in SFDA (Huang et al., 2021; Xia et al., 2021; Wang et al., 2022b; Yang et al., 2022; Zhao et al., 2022), TTA Chen et al. (2022), Domain Generalisation (DG) (Kim et al., 2021; Yao et al., 2022) and other studies (Vaze et al., 2022; Fei et al., 2022) to train/adapt a pre-trained model to a target domain using only unlabeled data. In previous works, contrastive learning has been used to get better feature representation. However, in this work, we align target features guided by source prototypes to get meaningful target features with respect to the source feature space. This results in meaningful clustering and good separation of classes in unseen domains.

## 3 Problem Setting and Motivation

In continual test-time domain adaptation, we are given a model trained using source domain data $\mathcal{D}_s = \{(x_i, y_i)\}_{i=1}^{n_s} \sim P_s$. Here, $(x_i, y_i)$ is source data and label which is drawn from the source distribution $P_s$. During testing, the source data is usually not available due to privacy concerns or storage constraints. At

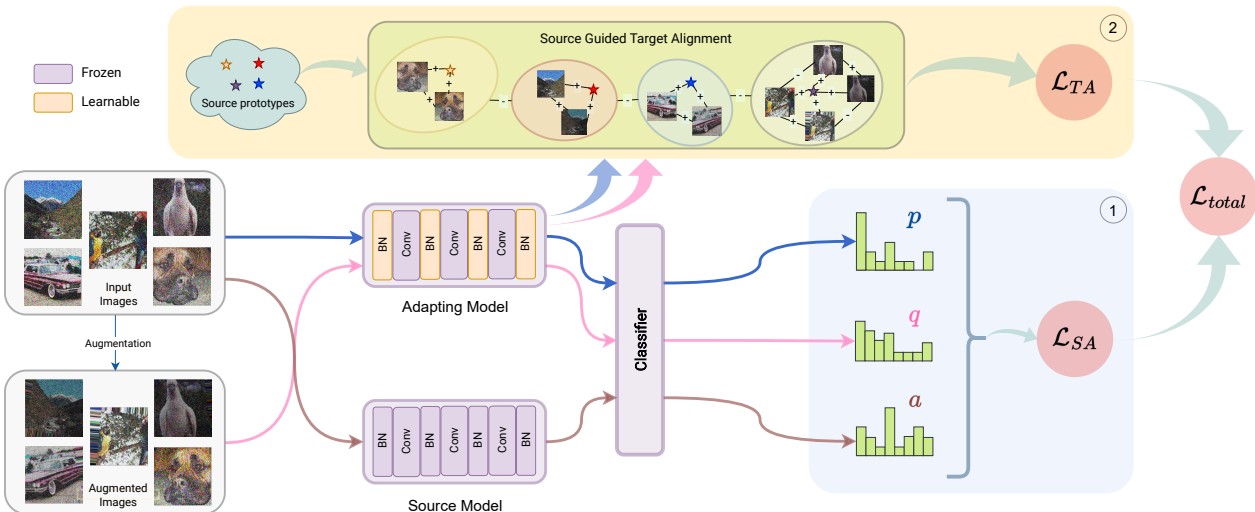

Figure 1: Illustration of the proposed SANTA framework. The original image and its augmentation are passed through the adapting model (only BN affine parameters are updated) and the source model. ① The prediction of the input given by the source model is used as an anchor (**a**) for the prediction given by the adapting model (**p** for input image and **q** for augmented image) for computing the source-anchoring loss $\mathcal{L}_{SA}$. ② The source guided target alignment $\mathcal{L}_{TA}$ uses the source prototypes to help maintain the semantic information learnt using the source domain and enforces clustering that is meaningful in the feature space.

this stage, the model encounters test data $\mathcal{D}_t = \{x_j\}_{j=1}^{n_t} \sim P_t$. In practice, the test data can belong to a different domain compared to the source, i.e. $P_t \neq P_s$. Further, $P_t$ can change over time such that $P_t^{(1)} \neq P_t^{(2)} \neq \ldots \neq P_s$ leading to our continual test time adaptation scenario.

In this work, the goal is to develop a practically useful framework for the above continual fully test time scenario, which should have the following characteristics: 1) For online adaptation, to reduce the latency, the model should be able to work seamlessly for different (preferably small) batch sizes, like 25 or 10 as used in (Zhao et al., 2023; Lim et al., 2023). But the current approaches Wang et al. (2022a) have been tested only for much larger batch sizes (like 200); 2) The updated model should continue to work well on the source domain; For example, while adapting to cloudy or rainy weather, the framework deployed for autonomous driving should not fail for images taken in clear weather (source domain); 3) In test-time adaptation scenario, since validation sets are usually not available for tuning hyper-parameters, the approach should require minimal hyper-parameters; and as a practical bonus, storage should be minimal and the framework should have low inference time for ease of deployment. Now, we describe the proposed SANTA framework (Figure 1), and elaborate on how some of these challenges are addressed.

## 4 Proposed SANTA Framework

Given the model trained using the source data, the goal is to adapt it using the limited amount of test-data encountered in each batch in a dynamic environment, while satisfying the desirable criteria mentioned above. Towards this goal, we propose (i) **S**ource **A**nchorization **N**etwork and (ii) **T**arget **A**lignment, which we now describe in detail.

### 4.1 Source Anchorization Network

During test-time, the model has access to few test samples in a batch, which may not be representative of the corresponding target distribution. Thus, modifying the model parameters completely on the basis of the available target data may result in simultaneously overfitting on the few target samples and also catastrophic forgetting of the source information. CoTTA Wang et al. (2022a) addressed this challenge by

(i) learning a teacher model by combining the source and a continuously adapting student model, wherein the teacher changes gradually for robust prediction and also using (ii) stochastic restoration to reset some of the model parameters to the source model after every batch. Though this gives impressive performance, it has two limitations, namely (i) the complete teacher, student and source model needs to be stored and (ii) the hyperparameters required for computing the teacher model and for stochastic restoration need to be determined, which can have different optimal values for different datasets. To overcome these challenges, in this work, we use self-distillation using the source model as the anchor (Hinton et al., 2015; Yang et al., 2019).

We denote the adapting model as $f_\theta$, as this is the model which is continuously updated and is used to predict the target data. The weights $\theta$ of the adapting model is initialised to the weights $\theta_s$ of the off-the-shelf source model $f_{\theta_s}$. Now, consider time instant $k$, when the model encounters a new batch denoted by $B_k$. Since adaptating the BN statistics has proven to be effective in capturing the data distribution characteristics (Schneider et al., 2020; Wang et al., 2022a), the BN statistics of the source model ($f_{\theta_s}$) and the adapting model ($f_\theta$) are changed to the BN statistics of the target batch at each step. Let these models be denoted as $f_{\theta_s}^k$ and $f_\theta^k$ respectively. $f_{\theta_s}^k$ (referred to as target corrected source (TCS) from now) can be thought of as a specialized model that accounts for the domain difference between the source and the specific target batch, On the other hand, the weights $\theta$ of the adapting model $f_\theta$ are optimized after every batch using the loss function that will be described later. It should also be noted that during optimization, we only update the BN parameters of the adapting model Wang et al. (2021).

The proposed self-distillation loss is inspired from the knowledge distillation loss formulation used in incremental learning (Dong et al., 2021; Kang et al., 2022) to prevent catastrophic forgetting. In this work, self-distillation between the response of the TCS model $f_{\theta_s}^k$ and the adapting model $f_\theta^k$ acts as a regularizer Mobahi et al. (2020), that encourages the adapting model to mimic the TCS model, which is a specialized model whose response corresponds to domain invariant features. Thus, our adapting model is reinforced to learn domain invariant features Mobahi et al. (2020), leading to better generalization, which we empirically show in Table 7.

The loss function is based on the prediction scores of the adapting model and the TCS model for a given batch of test images, $B_k := \{x_1, x_2, \ldots, x_N\}$, $N$ being the batch size. Let $p_{ij}$ and $a_{ij}$ denote the $j^{th}$ element of $f_\theta^k(x_i)$ (adapting model) and $f_{\theta_s}^k(x_i)$ (TCS model) respectively, which gives the prediction score of the $j^{th}$ class for the $i^{th}$ test image. The source-anchoring loss for a given batch is calculated as follows:

$$\mathcal{L}'_{\text{SA}} = -\frac{1}{N} \sum_{i=1}^{N} \sum_{j=1}^{C} p_{ij} \log(a_{ij}) \tag{1}$$

Here, $C$ is the number of classes, and $N$ is the batch size. Empirically, we observe that using augmentations of the test images make the model more robust. Let $q_{ij}$ denote the prediction score of the $j^{th}$ class for the $i^{th}$ augmented test image, given by the adapting model. The complete source-anchoring loss is given by

$$\mathcal{L}_{\text{SA}} = -\frac{1}{N} \sum_{i=1}^{N} \sum_{j=1}^{C} (p_{ij} \log(a_{ij}) + q_{ij} \log(a_{ij})) \tag{2}$$

This simple, yet effective self-distillation offers multiple advantages as follows: i) Since the target corrected source model is used for anchoring, there is no hyper-parameter involved (such as the weight for combining student with source to form the teacher model as in CoTTA Wang et al. (2022a)); ii) The adapting model can be directly used for prediction continuously, without requiring any restoration to the source model; iii) Since only the BN parameters are updated, just these parameters of the source need to be stored, resulting in much lesser storage requirements compared to storing two different models.

## 4.2 Target Alignment

The goal is to learn a generalized model as it encounters data from different domains, thereby making the features gradually domain invariant. Here, we additionally use self-supervision (in the form of contrastive

learning) for improved generalization as used in (Zhao et al., 2022; Wang et al., 2022b). Formally, the adapting model $f_\theta^k$ can be decomposed into a feature extractor, $g_\phi^k$ and the fixed classifier $h$, i.e.

$$f_\theta^k = h \circ g_\phi^k \tag{3}$$

Suppose, the augmented samples for the test batch $\{x_i\}_{i=1}^{N_k}$ are denoted by $\{x_i\}_{i=N_k+1}^{2N_k}$. We pass these samples through the feature extractor $g_\phi^k$ and further a projection head $p_\psi$ and normalized so that the features are mapped to a d-dimensional hyper-sphere (Chen et al., 2020; Khosla et al., 2020).

$$z_i = p_\psi \circ g_\phi^k(x_i) \tag{4}$$

The following is the contrastive loss

$$\mathcal{L}_{con} = \frac{1}{N} \sum_{i=1}^{N} \log \left( \frac{\exp(z_i.z_{N+i}/\tau)}{\sum_{k=1}^{2N} \mathbb{1}_{[k \neq i]} \exp(z_i.z_k/\tau)} \right) \tag{5}$$

where $\tau > 0$ is the temperature hyperparameter. In this work, as in Wang et al. (2021), the learnable model parameters $\phi$ comprise of only the BN affine parameters. The classifier is kept fixed throughout the adaptation process to avoid overfitting on the target samples and preserve the discriminative information learnt using the abundant source data. Thus, for correct classification, the target clusters should also align with the original source representations. To achieve this, we use the source prototype guided target alignment loss. To this end, we include a third view which assigns the nearest source prototype as an augmented view for the test time features. Specifically, let the source prototypes for the $C$ classes be denoted as $\{\pi_i\}_{c=1}^{C}$. We compute the nearest source prototype for a sample $x_i$ as

$$\pi(x_i) = \{\pi_t | t = \arg\max_c (\cos(\pi_c, g_\phi^k(x_i)))\} \tag{6}$$

For each feature $z_i$, we use two positives for contrastive learning. The first view $z_{N+i}$ is the projection of its augmented sample $x_{N+i}$, the second view $z_{2N+i}$ is the projection of its nearest class prototype $\pi(x_i)$. The source prototype guided Target Alignment loss is given by:

$$\mathcal{L}_{\text{TA}} = \frac{1}{2N} \sum_{i=1}^{N} \log \left( \frac{\exp(z_i.z_{N+i}/\tau)}{\sum_{k=1}^{3N} \mathbb{1}_{[k \neq i]} \exp(z_i.z_k/\tau)} \right) + \log \left( \frac{\exp(z_i.z_{2N+i}/\tau)}{\sum_{k=1}^{3N} \mathbb{1}_{[k \neq i]} \exp(z_i.z_k/\tau)} \right) \tag{7}$$

By using this additional view, we implicitly pass the class information to result in improved clustering while also keeping intact the semantic information learnt from the source data. The qualitative effect of clustering using prototypes as a view can be seen in Figure 2. We further quantitatively ablate its effect in Table 2.

### 4.3 Final loss

Given any off-the-shelf pre-trained model and the source prototypes, during testing, using the current test batch, we modify the BN parameters $\phi$ and the projection head $\psi$ such that the following loss is minimized:

$$\mathcal{L}_{\text{SANTA}} = \mathcal{L}_{\text{SA}} + \mathcal{L}_{\text{TA}} \tag{8}$$

The adapting model is used for predicting the class of all the test samples. It is robust enough to be updated continuously without any restoration back to the source model.

## 5 Experimental Evaluation

Here we describe the experiments performed to evaluate the effectiveness of the proposed framework.

**Dataset Details**:  Here, we evaluate the proposed framework extensively on multiple benchmark

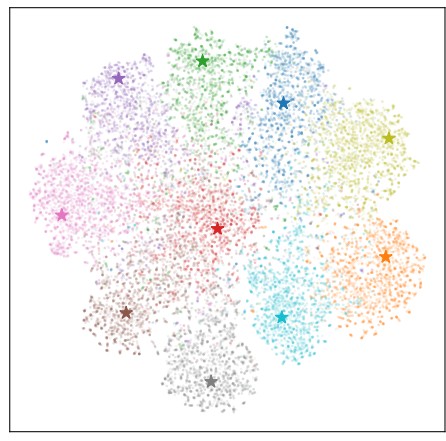
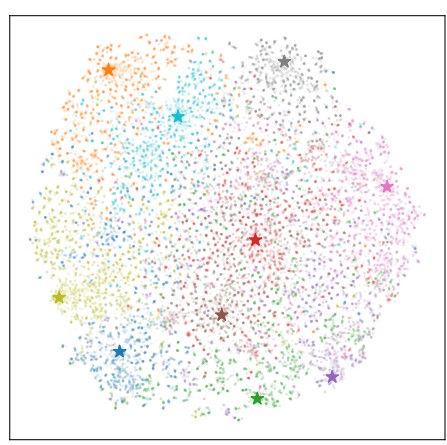

(a) With Source prototypes

(b) Without source prototypes

Figure 2: t-SNE plot of the feature space using only $\mathcal{L}_{TA}$ for adaptation. We observe that the alignment and clustering is better with source prototypes (a) as compared to not using them (b). The source prototypes are represented using stars.

| Dataset | Method | gaussian | shot | impulse | defocus | glass | motion | zoom | snow | frost | fog | brightness | contrast | elastic | pixelate | jpeg | Mean |
|---|---|---|---|---|---|---|---|---|---|---|---|---|---|---|---|---|---|
| CIFAR-10C | Source | 72.3 | 65.7 | 72.9 | 46.9 | 54.3 | 34.8 | 42.0 | 25.1 | 41.3 | 26.0 | 9.3 | 46.7 | 26.6 | 58.5 | 30.3 | 43.5 |
| | BN Stats Adapt | 28.1 | 26.1 | 36.3 | 12.8 | 35.3 | 14.2 | 12.1 | 17.3 | 17.4 | 15.3 | 8.4 | 12.6 | 23.8 | 19.7 | 27.3 | 20.4 |
| | TENT-continual | 24.8 | 20.6 | 28.6 | 14.4 | 31.1 | 16.5 | 14.1 | 19.1 | 18.6 | 18.6 | 12.2 | 20.3 | 25.7 | 20.8 | 24.9 | 20.7 |
| | CoTTA | 24.3 | 21.3 | 26.6 | 11.6 | 27.6 | 12.2 | 10.3 | 14.8 | 14.1 | 12.4 | 7.5 | 10.6 | 18.3 | 13.4 | 17.3 | 16.2 |
| | RMT | 24.5 | 20.0 | 25.5 | 13.9 | 24.6 | 14.9 | 13.3 | 16.0 | 15.8 | 15.6 | 11.1 | 15.0 | 18.3 | 14.6 | 16.9 | 17.3 |
| | PETAL | 23.4 | 21.1 | 25.7 | 11.7 | 27.2 | 12.2 | 10.3 | 14.8 | 13.9 | 12.6 | 7.4 | 10.5 | 18.1 | 13.4 | 16.8 | **15.95** |
| | **SANTA** | 23.9 | 20.1 | 28.0 | 11.6 | 27.4 | 12.6 | 10.2 | 14.1 | 13.2 | 12.2 | 7.4 | 10.3 | 19.1 | 13.3 | 18.5 | 16.1 ± 0.06 |
| CIFAR-100C | Source | 73.0 | 68.0 | 39.4 | 29.3 | 54.1 | 30.8 | 28.8 | 39.5 | 45.8 | 50.3 | 29.5 | 55.1 | 37.2 | 74.7 | 41.2 | 46.4 |
| | BN Stats Adapt | 42.1 | 40.7 | 42.7 | 27.6 | 41.9 | 29.7 | 27.9 | 34.9 | 35.0 | 41.5 | 26.5 | 30.3 | 35.7 | 32.9 | 41.2 | 35.4 |
| | TENT-continual | 37.2 | 35.8 | 41.7 | 37.9 | 51.2 | 48.3 | 48.5 | 53.7 | 63.7 | 71.1 | 70.4 | 82.3 | 88.0 | 88.5 | 90.4 | 60.9 |
| | CoTTA | 40.1 | 37.7 | 39.7 | 26.9 | 38.0 | 27.9 | 26.4 | 32.8 | 31.8 | 40.3 | 24.7 | 26.9 | 32.5 | 28.3 | 33.5 | 32.5 |
| | RMT | 40.5 | 36.1 | 36.3 | 27.7 | 33.9 | 28.5 | 26.4 | 29.0 | 29.0 | 32.5 | 25.1 | 27.4 | 28.2 | 26.3 | 29.3 | 30.4 |
| | PETAL | 38.3 | 36.4 | 38.6 | 25.9 | 36.7 | 27.2 | 25.4 | 32.0 | 30.8 | 38.7 | 24.4 | 26.4 | 31.5 | 26.9 | 32.5 | 31.46 |
| | **SANTA** | 36.5 | 33.1 | 35.1 | 25.9 | 34.9 | 27.7 | 25.4 | 29.5 | 29.9 | 33.1 | 23.6 | 26.7 | 31.9 | 27.5 | 35.2 | **30.3** ± 0.05 |
| ImageNet-C(5k) | Source | 97.8 | 97.1 | 98.2 | 81.7 | 89.8 | 85.2 | 78.0 | 83.5 | 77.1 | 75.9 | 41.3 | 94.5 | 82.5 | 79.3 | 68.6 | 82 |
| | BN Stats Adapt | 85.0 | 83.7 | 85.0 | 84.7 | 84.3 | 73.7 | 61.2 | 66.0 | 68.2 | 52.1 | 34.9 | 82.7 | 55.9 | 51.3 | 59.8 | 68.6 |
| | TENT-continual | 81.6 | 74.6 | 72.7 | 77.6 | 73.8 | 65.5 | 55.3 | 61.6 | 63.0 | 51.7 | 38.2 | 72.1 | 50.8 | 47.4 | 53.3 | 62.6 |
| | CoTTA | 84.7 | 82.1 | 80.6 | 81.3 | 79.0 | 68.6 | 57.5 | 60.3 | 60.5 | 48.3 | 36.6 | 66.1 | 47.2 | 41.2 | 46.0 | 62.7 |
| | RMT | 80.2 | 76.4 | 74.5 | 77.1 | 74.4 | 66.2 | 57.6 | 57.0 | 59.1 | 48.0 | 39.1 | 60.6 | 47.3 | 42.5 | 43.4 | 60.2 |
| | **SANTA** | 74.1 | 72.9 | 71.6 | 75.7 | 74.1 | 64.2 | 55.5 | 55.6 | 62.9 | 46.6 | 36.1 | 69.9 | 50.6 | 44.3 | 48.5 | **60.1** ± 0.06 |
| ImageNet-C(50k) | Source | 97.8 | 97.1 | 98.1 | 82.1 | 90.2 | 85.2 | 77.5 | 83.1 | 76.7 | 75.6 | 41.1 | 94.6 | 83.0 | 79.4 | 68.4 | 82.0 |
| | BN Stats Adapt | 84.9 | 84.0 | 84.2 | 85.0 | 84.7 | 73.6 | 61.2 | 65.6 | 66.9 | 52.0 | 34.7 | 83.2 | 55.8 | 51.0 | 60.2 | 68.5 |
| | TENT-continual | 71.5 | 66.1 | 69.3 | 82.3 | 90.0 | 94.9 | 97.0 | 98.8 | 99.3 | 99.3 | 99.2 | 99.6 | 99.4 | 99.4 | 99.4 | 91.0 |
| | CoTTA | 78.4 | 68.4 | 64.4 | 74.8 | 71.8 | 69.3 | 67.4 | 72.1 | 71.1 | 67.0 | 62.2 | 73.5 | 69.4 | 67.1 | 68.6 | 69.7 |
| | RMT | 73.6 | 65.9 | 64.3 | 74.3 | 72.0 | 71.0 | 69.9 | 70.2 | 71.9 | 70.3 | 66.2 | 74.7 | 68.5 | 67.3 | 67.9 | 69.9 |
| | **SANTA** | 73.6 | 75.1 | 73.2 | 76.2 | 76.8 | 64.1 | 53.5 | 55.8 | 61.7 | 43.7 | 34.5 | 72.7 | 49.2 | 43.9 | 50.2 | **60.3** ± 0.07 |

Table 1: Error percentages (lower is better) of Source model, BN Stats Adapt Schneider et al. (2020), TENT-Continual Wang et al. (2021), CoTTA Wang et al. (2022a), RMT Döbler et al. (2023), PETAL Brahma & Rai (2023) for CIFAR-10C, CIFAR-100C and ImageNet-C (5k and 50k samples) for batchsizes of 200, 200 and 64 respectively. For SANTA, the standard deviation is reported over 5 random seeds.

datasets, namely, CIFAR-10C, CIFAR-100C and ImageNet-C Hendrycks & Dietterich (2019). These datasets have 10, 100 and 1000 classes respectively. All the datasets contain 15 diverse corruptions of the

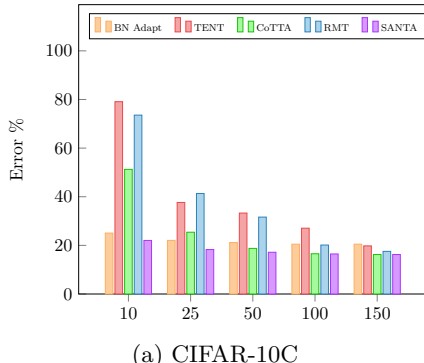 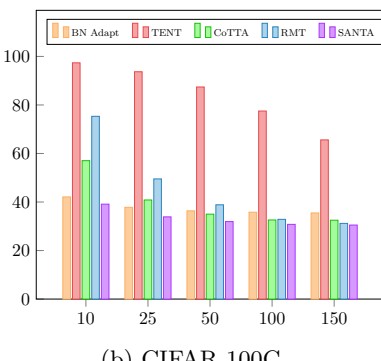 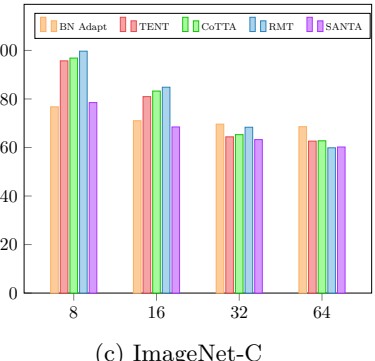

(a) CIFAR-10C     (b) CIFAR-100C     (c) ImageNet-C

Figure 3: These figures compare the robustness of the different approaches for different batch sizes. The x-axis is the batch size and y-axis is the error percentage (lower is better). We observe that the proposed SANTA is robust across batch sizes.

| Methods | CIFAR-10C | | | | | | CIFAR-100C | | | | | | IMAGENET-C | | | |
|---|---|---|---|---|---|---|---|---|---|---|---|---|---|---|---|---|
| | 200 | 150 | 100 | 50 | 25 | 10 | 200 | 150 | 100 | 50 | 25 | 10 | 64 | 32 | 16 | 8 |
| SALoss (original image) | 20.1 | 20.1 | 20.4 | 20.7 | 21.7 | 24.8 | 32.6 | 32.8 | 33.2 | 34.0 | 35.7 | 40.7 | 62.7 | 64.9 | 69.4 | 77.8 |
| SALoss (original + augmented image) | 17.1 | 17.2 | 17.4 | 18.0 | 19.0 | 22.5 | 31.4 | 31.6 | 32.0 | 32.8 | 34.7 | 39.9 | 61.3 | 63.8 | 68.5 | **76.8** |
| **SANTA** (SALoss + TALoss) | **16.1** | **16.2** | **16.5** | **17.2** | **18.3** | **22.0** | **30.4** | **30.5** | **30.8** | **31.9** | **33.9** | **39.1** | **60.2** | **63.2** | **68.5** | 78.5 |

Table 2: Mean error percentage (lower is better) demonstrating the importance of the two proposed components. Ablation is done on all the four datasets for multiple batch sizes.

forms noise, blur, weather, and digital. Each corruption has five levels of severity, applied to the test set of all the three datasets. Following CoTTA Wang et al. (2022a), we do ImageNet-C experiments with 5k samples per corruption. Further, we do a more robust experiment using the complete ImageNet-C dataset consisting of 50k samples for each corruption*. For all the experiments, unless mentioned otherwise, the test sequence consists of all 15 corruptions at the highest level of severity Wang et al. (2022a). The goal is to adapt an off-the-shelf source model to this dynamically changing environment efficiently during test time.

**Implementation Details**: For CIFAR-10C, we use a pre-trained WideResNet-28 Zagoruyko & Komodakis (2016) model from the RobustBench benchmark Croce et al. (2021) as in Wang et al. (2022a). The model is updated with one gradient step per batch, and the Adam optimizer with a learning rate of 1e-3 is used. The temperature is set to the default value of 0.1. The CIFAR-100C experiment uses a pre-trained ResNeXt-29 Xie et al. (2017) model, which is one of the default models for CIFAR-100 in the RobustBench benchmark Croce et al. (2021). The same hyperparameters as the CIFAR-10 experiment are used. For the ImageNet-C experiment, the standard pre-trained Resnet50 He et al. (2016) model from RobustBench Croce et al. (2021) is used. Here, SGD is used as the optimiser with a learning rate of 1e-2 as in Wang et al. (2022a). We conduct all the experiments on an NVIDIA GeForce RTX 3090.

## 6 Research Questions

We perform experiments to address the following research questions:

**1) How does SANTA perform on the standard experimental protocol (higher batch size) as used in Wang et al. (2022a)?**
Table 1 reports the results on the three benckmark datasets. The batch sizes used for these experiments are 200, 200 and 64 for CIFAR-10C, CIFAR-100C and ImageNet-C respectively as in Wang et al. (2022a). In the TENT-continual setup, the model continuously adapts and is not reset to the source model after each corruption.

---

*Previous works have used only 5k samples per corruption for ImageNet-C due to a bug in Robustbench setup.

| Avg. Error (%) | Source | BN Stats Adapt | TENT-Continual | CoTTA | RMT | SANTA |
|---|---|---|---|---|---|---|
| ImageNet-C | 82.0 | 68.5 | 84.1 ± 6.38 | 64.61 ± 2.72 | 63.6 ± 4.15 | 60.18 ± 0.13 |

Table 3: Average error for ImageNet-C experiments over 10 diverse corruption sequences (severity level 5).

| | CIFAR-10C | CIFAR-100C | ImageNet-C |
|---|---|---|---|
| SANTA w/ prototypes | 16.1 | 30.0 | 60.2 |
| SANTA w/o prototypes | 16.2 (-0.1) | 31.4 (-1.4) | 61.1 (-0.9) |

Table 4: Here, we see the performance comparison (error % - lower is better) between using prototypes as a view v/s not using them in the target alignment loss. (.) is the relative drop in performance.

We observe, from Table 1 that for all the datasets, the proposed method SANTA consistently performs at par or surpasses the existing approaches. For ImageNet-C, using 5k samples per corruption, we obtain an error of 60.1%, which is 2.6% better than the seminal work CoTTA Wang et al. (2022a) and at par with RMT Döbler et al. (2023). Moreover, in the challenging case of ImageNet-C, using all 50k samples, SANTA significantly outperforms existing baselines. No hyperparameter tuning is done for any methods on extending the protocol to 50k samples. This is in congruence with the fact that, in real world CTTA, tuning hyperparameters is out of scope.

**2) How does SANTA perform on lower batch sizes, which is more realistic scenario for online test time adaptation?**
A practical test-time adaptation algorithm should work satisfactorily for lower batch sizes, which will decrease the average time for inference of a sample and thus lower the latency of the framework. Although researchers have recently started to address the issue of robustness across batch sizes (Lim et al., 2023; Gong et al., 2022), these are not fully test time adaptation and requires source data for warmup before online adaptation.

Here, we evaluate the robustness of the proposed SANTA framework for lower batch sizes and compare the results with the state-of-the-art. Figure 3 shows the results for the three datasets with decreasing batch sizes. Since the results of the other approaches were not reported for other batch sizes, we used the official codes and obtained the results reported in the table. To ensure the best results for the other methods, we tuned the appropriate hyperparameters. Specifically, for TENT and RMT, we varied the learning rate. For CoTTA, the restoration probability and model EMA factor was reduced in proportion to the decrease in batch size. Note that for the proposed SANTA, *no parameters were changed for a dataset across batch sizes*. We observe that the improvement provided by our approach becomes clearer as the batch size decreases. For example, for a batch size of 10, we obtain 22.0% for CIFAR-10C dataset, which is significantly better compared to the next best of 25.0% obtained by BN Stats Adapt as shown in Figure 3. This experiment justifies the effectiveness of the proposed SANTA for online settings. Also, prior methods RMT and CoTTA perform significantly worse compared to SANTA as the batch size decreases, suggesting that it can be unreliable to use them in real-world settings where more often than not there is both less data and therefore less batch size from a particular domain.

**3) Are both the proposed modules important?**
Here, we analyze the importance of the two losses in the proposed SANTA framework. We observe from Table 2 that a significant portion of the performance improvement of SANTA can be attributed to the source-anchoring of the test samples and its augmented version. The clustering and alignment terms further help to improve the performance, thereby achieving state-of-the-art performance for continual test-time domain adaptation under different challenging settings.

**4) How does SANTA perform for different corruption orders?**
From Table 3, we observe that SANTA demonstrates competitive performance on par with RMT and notably outperforms all previously employed methods. Additionally, it is observed that SANTA exhibits low variance across corruptions, compared to other techniques, which is a highly sought-after characteristic for ensuring the reliability of CTTA.

| Avg. Error (%) | Source | BN Stats Adapt | TENT-continual | CoTTA | RMT | SANTA |
|---|---|---|---|---|---|---|
| CIFAR-10C | 24.8 | 13.7 | 20.4 | 10.9 | **10.4** | 10.7 |
| CIFAR-100C | 33.6 | 29.9 | 74.8 | 26.3 | 26.6 | **25.6** |

Table 5: Average error over all corruptions with severity presented in the gradual test time adaptation manner. The order of corruption used here is the same as in Table 1.

| batchsize → | 200 | 150 | 100 | 50 | 25 | 10 |
|---|---|---|---|---|---|---|
| TENT-Continual | 92.0 (-70.9) | 96.2 (-75.1) | 97.8 (-76.7) | 98.2 (-77.1) | 98.5 (-77.4) | 99.0 (-77.9) |
| CoTTA | 22.8 (-1.7) | 23.0 (-1.9) | 23.9 (-2.8) | 27.2 (-6.1) | 35.5 (-14.4) | 58.5 (-37.4) |
| RMT | 24.7 (-3.6) | 26.3 (-5.2) | 29.0 (-8.9) | 35.9 (-14.8) | 48.0 (-26.9) | 76.1 (-55.0) |
| SANTA | **22.4 (-1.3)** | **22.7 (-1.6)** | **22.9 (-1.8)** | **23.6 (-2.5)** | **25.8 (-4.7)** | **29.9 (-8.8)** |

Table 6: Error on CIFAR-100 test set after the model has been adapted to all the corruptions as in Table 1. (.) is the degradation in performance on source data compared to the source model, whose error is **21.1**%. This deviation quantifies the amount of catastrophic forgetting while adaptation.

**5) What is the effect of source prototypes in Target Alignment?**
We perform experiments with and without source prototypes to study its effect and report the results in Table 4. We observe that without the prototypes, the performance drops slightly compared to when using it. Although source prototypes aid clustering as seen in Figure2, our method does not primarily rely on it, suggesting that SANTA can still be reliably used in cases where source prototypes are not accessible.

**6) How does SANTA perform when the domain shifts are gradual?**
In the standard setup, the corruption types change abruptly with maximum severity levels. A more realistic approach will be to evaluate the performance of the approaches when the severity levels change gradually over a sequence of 15 corruption types. Thus we experiment with the gradual setup as also done in Wang et al. (2022a). The representation below shows the order in which severity is faced for every corruption.

$$\underbrace{\ldots \to 2 \to 1}_{t\text{-}1 \text{ and before}} \to \underbrace{1 \to 2 \to 3 \to 4 \to 5 \to 4 \to 3 \to 2 \to 1}_{t \text{ corruption type, with changing severity}} \to \underbrace{1 \to 2 \to \ldots}_{t+1 \text{ and after}}$$

The results suggest that BN Adapt, CoTTA, RMT and SANTA are more effective than the source and TENT-continual approaches in dealing with corruption types that change gradually over time. For CIFAR-10C, we observe CoTTA, RMT and SANTA perform similarly. For CIFAR-100C, our approach achieves the lowest average error rate of 25.6%, followed by CoTTA with an average error rate of 26.3%. BN Stats Adapt approach also performs well with an average error rate of 29.9%.

**7) Does SANTA prevent catastrophic forgetting by retaining good performance on revisiting the source data?**
Although the source trained model is gradually adapted to the changing testing conditions, we also want it to be able to perform well on the original source distribution, which requires the model to not catastrophically forget the original source training information. For example, the original model trained on clear weather conditions should continue to do well for clear weather images, even though it has adapted to other conditions like rainy, foggy, etc. To achieve these contrasting goals, it is important that the model has the right balance of stability-plasticity. The stability-plasticity trade-off refers to the balance between preserving learned knowledge and adapting to new information. Table 6 reports the results of using the model adapted on CIFAR-100C, on a held-out testing set from the source distribution, which is the clean CIFAR-100 dataset in this case. The performance of the original model trained on source gives an error of 21.1% on its test set. Thus the difference gives an estimate of the catastrophic forgetting (Table 6). We observe that even after adaptation, SANTA is able to maintain its performance on the source distribution very well as compared to CoTTA, RMT and TENT for all batch sizes.

| Method | snow | frost | fog | brightness | contrast | elastic | pixelate | jpeg | Mean |
|---|---|---|---|---|---|---|---|---|---|
| BN Stats Adapt | 35.6 | 34.9 | 42.1 | 26.9 | 31.0 | 36.0 | 33.5 | 41.7 | 35.2 |
| CoTTA | 31.3 | 30.5 | 36.7 | 25.2 | 27.8 | 31.5 | 29.1 | 36.4 | **31.1** |
| RMT | 32.1 | 32.1 | 38.6 | 28.1 | 33.0 | 30.8 | 28.7 | 34.3 | 32.2 |
| SANTA | 31.2 | 31.2 | 36.3 | 25.0 | 28.7 | 32.2 | 28.0 | 37.2 | 31.2 |

Table 7: This experiment on CIFAR-100C demonstrates the generalizability of the learnt model. We observe that both CoTTA and SANTA yield a more generalized model after adaptation.

| Method | # Parameters | # Trainable | % Trainable |
|---|---|---|---|
| BN-Stats | 6,900,132 | 0 | 0 |
| TENT | 6,900,132 | 25,216 | 0.365 |
| CoTTA | 20,700,396 | 6,900,132 | 33.333 |
| RMT | 13,947,976 | 6,900,132 | 50.000 |
| SANTA | 7,047,972 | 172,928 | 2.506 |

Table 8: Number of (trainable) parameters as a proxy for the storage requirement of the respective algorithms. This table is for CIFAR-100C with ResNeXt-29 as the backbone.

**8) How well does the model generalize?**
When the source model encounters data from different domains, we want it to gradually become more generalized, such that the features become domain invariant. To evaluate whether this happens in practice, we perform an experiment, in which the model is first adapted on the first 7 corruptions (gaussian to zoom). We then freeze the weights and evaluate its performance on the last 8 corruptions (snow to jpeg). We compare the performance of this adapted model (using the methods CoTTA and SANTA) to the performance using the optimization free BN Stats Adapt model, where BN statistics are adapted to the corresponding batch statistics. We see from Table 7 that both CoTTA and the proposed framework have indeed learnt more generalized features and is therefore performing better than the BN Stats Adapt model. SANTA is however, a much lighter framework when compared to CoTTA, which we discuss in detail below.

**9) How does the model fare in terms of storage cost, inference time?**
The proposed SANTA framework has computational advantages over the existing approaches:

(i) Number of parameters to be stored: Table 8 reports the total number of parameters and trainable parameters for TENT, CoTTA, RMT and SANTA. If the source backbone has $N$ parameters, CoTTA stores $3N$ parameters (source, teacher and student). RMT stores 2N parameters(teacher and student). We only store $1.02N$ parameters (source, BN parameters of the adapting model which is only 2% of N). Thus the proposed framework is much simpler and computationally efficient, which can be especially important in real-time applications where time and computational resources are limited. Additionally, as we freeze the convolution blocks and only optimize the BN affine parameters in SANTA, it is less prone to catastrophic forgetting as it is less likely to overfit to the test time data. This is important in test-time adaptation scenarios where the model must adapt to new data quickly and accurately.

(ii) Inference time for a fixed batch size: The inference time of a model is an important factor in test-time adaptation because it determines how quickly the model can be applied to new data. In Figure 4 we have compared the inference time (in sec) per batch (of 200 samples) for RMT, CoTTA and SANTA. SANTA does 2 forward and 1 backward pass per batch which takes 0.224 sec/batch. RMT does 2 forward, 1 backward pass and model EMA update, which takes 0.26 sec/batch. On the other hand, CoTTA takes 2.2 sec/batch on average as it does 32 forward passes in the worst case and 1 backward pass, implying that SANTA is ~10X faster than CoTTA. This can be attributed to the fact that CoTTA uses 32 forward passes in the worst case

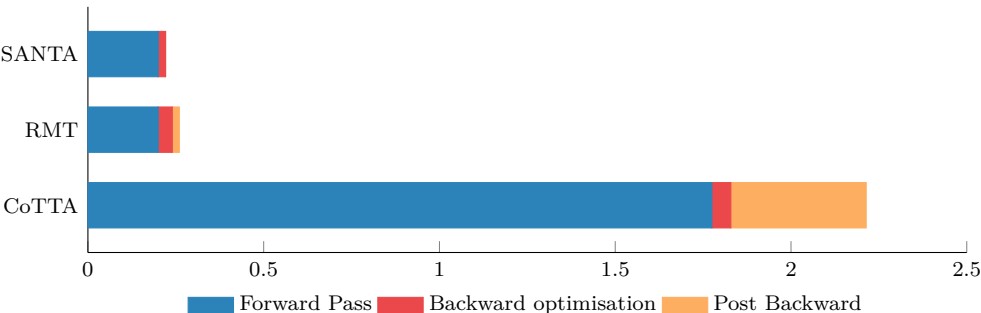

Figure 4: Comparison of inference time of the proposed SANTA framework with RMT and CoTTA. The x-axis is time (wall clock time) of inference per batch (sec/batch). These experiments were done on CIFAR-100C using NVIDIA GeForce RTX 3090.

for its prediction and also updates the teacher model after every step. Although RMT and SANTA are both computationally efficient compared to CoTTA, our method has a clear advantage in low batch size setting as it significantly outperforms RMT, and hence is more effective in varied settings.

## 7 Conclusion

In this paper, we proposed a novel SANTA framework for the challenging task of continual test-time domain adaptation. The proposed approach modifies the batch-norm affine parameters using source anchoring-based self-distillation to ensure the model incorporates knowledge of newly encountered domains while avoiding catastrophic forgetting. Additionally, source prototype guided target alignment is proposed to maintain the already learned semantic information while grouping target samples naturally. The approach is quite robust to decreasing batch sizes, justifying its effectiveness for online application. The SANTA framework offers additional advantages such as retaining performance on the source domain, having minimal tunable hyper-parameters and storage requirements, in addition to achieving state-of-the-art results on all the benchmark datasets.

## 8 Acknowledgements

This work is partly supported through a research grant from SERB, Department of Science and Technology, Govt. of India (SPF/2021/000118). The authors would like to thank Jayateja Kalla for helpful discussions throughout this work.

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

# Appendix

## A  Experiment on CCC

This benchmark was pointed out during the review process. The Continuously Changing Corruptions (CCC) Press et al. (2023) benchmark is generated from ImageNet-C by interpolating between domains in ImageNet-C to produce any number of samples. By default, CCC consists of 7.5 million samples which is $10\times$ more than the samples in ImageNet-C. This creates a benchmark which tracks the long term performance of CTTA frameworks. The CCC benchmark shows that prior CTTA methods fail across long adaptation time-scales. Even in this challenging long term setting, we observe that SANTA performs consistently better than prior methods as shown in Table 9.

| Method | CCC-Medium | CCC-Hard |
|--------|-----------|----------|
| BNStats | 27.9 $\pm$0.74 | 6.00 $\pm$0.31 |
| CoTTA | 7.70 $\pm$0.43 | 1.10 $\pm$0.16 |
| ETA | 1.1 $\pm$0.43 | 0.23 $\pm$0.05 |
| EATA | 35.4 $\pm$1.0 | 8.70 $\pm$0.80 |
| RDumb | 38.9 $\pm$1.4 | 9.60 $\pm$1.60 |
| SANTA | 32.7 $\pm$0.8 | 9.10 $\pm$0.60 |

Table 9: Percentage accuracy of CTTA methods (higher is better) on CCC benchmark.

## B  Ablation on Temperature

|  | 0.005 | 0.01 | 0.05 | 0.1 | 0.5 | 1.0 | 5.0 |
|--|-------|------|------|-----|-----|-----|-----|
| CIFAR10-C | 16.1 | 16.1 | 16.1 | 16.1 | 16.3 | 16.5 | 16.7 |
| CIFAR100-C | 30.3 | 30.3 | 30.3 | 30.3 | 30.8 | 31.2 | 31.3 |
| ImageNet-C | 60.6 | 60.4 | 60.3 | 60.2 | 61.0 | 61.1 | 61.2 |

Table 10: Ablation showing the sensitivity of our method on all considered datasets. The above numbers are error % (lower is better)

Table 10 shows the performance of the model for different temperature values. The best performance is approximately achieved for a temperature value of 0.1, indicating that a soft probability distribution is more suitable for this model. It is also worth noting that the model is fairly resilient to a large range of temperature values, implying that the model's performance does not deteriorate significantly for different temperature values. This suggests that the model is robust to variations in temperature hyperparameter and performs reasonably well for a wide range of values.

## C  Predict and Adapt Code

Here, we present the code for our method. This code is in the format of code provided by the CoTTAWang et al. (2022a) and TENTWang et al. (2021). The code for contrastive learning loss can be found here Khosla et al. (2020).

```python
def forward_and_adapt(x):
    imgs_test = x[0]

    optimizer.zero_grad()

    # forward original test data
```

```
features_test = feature_extractor(imgs_test)
outputs_test = classifier(features_test)

# forward augmented test data
features_aug_test = feature_extractor(tta_transform((imgs_test)))
outputs_aug_test = classifier(features_aug_test)

# forward original test data through the source model
outputs_src = model_src(imgs_test)

with torch.no_grad():
    # dist[:, i] contains the distance from
    # every source sample to one test sample
    dist = F.cosine_similarity(
    prototypes_src.repeat(
    1, features_test.shape[0], 1
    ),
    features_test.unsqueeze(0).repeat(
    prototypes_src.shape[0], 1, 1
    ), dim=-1
    )

    # for every test feature, get the nearest
    # source prototype and derive the label
    _, indices = dist.topk(1, largest=True, dim=0)
    indices = indices.squeeze(0)

# Here we see the 3-views that are passed to
# contrastive loss
features = torch.cat([prototypes_src[indices],
                    features_test.unsqueeze(1),
                    features_aug_test.unsqueeze(1)],
                    dim=1)

# Adapted from:
# https://github.com/HobbitLong/SupContrast/blob/master/losses.py
loss_contrastive = contrastive_loss(features=features)

loss_entropy = symmetric_entropy(x=outputs_test,
                                x_aug=outputs_aug_test,
                                x_src=outputs_src).mean(0)
loss_trg = loss_entropy + loss_contrastive
loss_trg.backward()
optimizer.step()

return outputs_test
```

We finally take the $\arg\max(\text{output\_test})$ across the sample dimension to get our final prediction for the target batch.

## D Augmentation

We use the same augmentations as used in CoTTA here, we explicitly report the series of random augmentations along with their ranges for better reproducibility.

```python
n_pixels = img_shape[0]
clip_min, clip_max = 0.0, 1.0
p_hflip = 0.5
gaussian_std = 0.005

transforms = transforms.Compose([
    Clip(0.0, 1.0),
    ColorJitterPro(
    brightness=[0.6, 1.4],
    contrast=[0.7, 1.3],
    saturation=[0.5, 1.5],
    hue=[-0.06, 0.06],
    gamma=[0.7, 1.3]
    ),
    transforms.Pad(padding=int(n_pixels / 2),
                    padding_mode='edge'),
    transforms.RandomAffine(
        degrees=[-15, 15],
        translate=(1/16, 1/16),
        scale=(0.9, 1.1),
        shear=None,
        resample=PIL.Image.BILINEAR,
        fillcolor=None
    ),
    transforms.GaussianBlur(kernel_size=5,
                            [0.001, 0.5]),
    transforms.CenterCrop(size=n_pixels),
    transforms.RandomHorizontalFlip(p=p_hflip),
    GaussianNoise(0, gaussian_std),
    Clip(clip_min, clip_max)
    ])
    return tta_transforms
```

