# OpenReview forum: "SANTA: Source Anchoring Network and Target Alignment for Continual Test Time Adaptation"
_TMLR — Accepted by TMLR_

### Review · Reviewer_noN9 · 2023-08-17

**Summary Of Contributions:**

The authors study the task of continuous test-time adaptation when the test distribution continuously changes over time. They propose a new test-time adaptation loss which consists of two parts. The first part is a source anchoring network which is supposed to prevent the model to divert too strongly from its initial condition. The second part of the loss is meant to increase the target alignment of the adapted model with a contrastive loss. The authors show results on CIFAR10-C, CIFAR100-C and ImageNet-C.

**Audience:**

Yes

**Broader Impact Concerns:**

The authors did not include a section on ethical implications of this work. They use publicly available datasets and models, so I do not have any specific ethical concerns for this work.

**Claims And Evidence:**

No

**Requested Changes:**

In CoTTA, the authors average the results across 10 corruption orders. Has this been done in this paper as well? If not, the experiments should be rerun.

There is a known bug if the CoTTA evaluation code is used, such that only 5k images are sampled instead of 50k: https://github.com/qinenergy/cotta/blob/main/imagenet/conf.py, line 55. The authors of robustbench are aware of this issue: https://github.com/RobustBench/robustbench/issues/92. Could the authors make sure that their evaluation does not suffer from this bug?

Please include ETA as baseline in your method. ETA does not include source samples as part of their method, so the reasoning that the authors cannot use this method as comparison does not hold. The authors will need to run this experiment as no ETA (lifelong) results were included in the ETA paper.

The recently released Continuously Changing Corruptions [F] benchmark has been proposed to examine the continuous test-time adaptation over long time scales. They find that all SotA TTA techniques eventually collapse to chance level or at least baseline performance. Since this approach is quite similar to CoTTA and CoTTA does collapse over time, could the authors please test on the CCC benchmark to test whether SANTA maintains performance over a longer time-scale? If the authors evaluate on CCC, comparing to RDumb, the method which was proposed in CCC would also be interesting.

[F] Press et al. “RDumb: A simple approach that questions our progress in continual test-time adaptation”

Please include the missing related works and respond to / change the text following my remarks in the Strengths and Weaknesses section above. Especially the paragraphs denoted by **major** are important. Please do a round of proof-reading to eliminate all typos.

The paper would benefit from a "Limitations" section.


**Strengths And Weaknesses:**

## Strengths:

The problem of continuous test-time adaptation is important.

The paper is generally well-written and easy to follow.

The approach is fairly clear and well-motivated.

The authors compare their method to several baselines, although they did miss some.

The Tables and Figures are well designed and easy to read.

## Weaknesses:
I have some questions with regards to the experiments which I formulated in the Requested Changes field more explicitly:

1)	It is not clear to me whether the authors used 10 corruption orderings as has been done in CoTTA.
2)	I wonder whether the evaluation might suffer from a known bug that robustbench only gives you 5k examples when loading ImageNet-C.
3)	It is not clear to me whether their method would collapse (just like CoTTA) does when evaluated on a longer time-scale.
4)	Some related works have been missed and should be included, see list below.
5)	Some interpretations of the results are wrong: The authors write that their method outperforms all other methods which is not true looking at the numbers in the Table.

In general, the results in this paper are very close to the baseline numbers the authors compare to. Their approach is much more computationally efficient which is nice. Thus, even if the results are on par with other SotA approaches, but the method is much more efficient, this would be fine. But then, the authors should change their wording, and not claim that their method “outperforms all other methods”. The wording that SANTA is an effective TTA method (used in the abstract + contributions) is fine.


### Related work:
The authors should include the following works in their TTA section:

- [A] Liu et al. “Ttt++: When does self-supervised test-time training fail or thrive?”
- [B] Eastwood et al. “Source-free adaptation to measurement shift via bottom-up feature restoration”
- [C] Mummadi et al. “Test-time adaptation to distribution shift by confidence maximization and input transformation”
- [D] Rusak et al. “If your data distribution shifts, use self-learning”
- [E] Bartler et al. “Mt3: Meta test-time training for self-supervised test-time adaption”
- [G] Zhang et al. "MEMO: Test Time Robustness via Adaptation and Augmentation"

I don’t think that any of these works consider continuous TTA, so it’s not necessary to include them as baselines.

### General:
Page 2: “The proposed SANTA framework also utilizes contrastive learning to ensure better model generalizability to unseen domains.” -> It is not clear how contrastive learning would help here. Not enough information for this claim, not well motivated. Please either provide references or expand upon this thought.

Page 2: “SFDA” has not been introduced as an abbreviation, please fix.

Table 1: I would include citations next to the methods to make it easier for the reader to jump to the references section.

[**major**]: “We observe that for all the datasets, the proposed SANTA outperforms all the other existing
approaches.” -> This is not true: PETAL outperforms SANTA on CIFAR10-C, and SANTA performs within 1 p.p. better than RMT on CIFAR100-C and ImageNet-C. I don’t think this is even significant.

Figure 3: I would add “Error rate” as the y-label to the left of the first subplot.

[**major**] Figure 3: The BN adapt numbers look strange. Schneider et al. show the performance of a RN50 depending on the batch size, and find that with a batch size of 8, they get more or less the baseline performance and with a batch size of 64, they get close to optimal performance. The difference in error rate should be something around 10-15 p.p., while in Fig. 3c, the performance looks fairly constant for the orange curve across the batch sizes.

### Typos:
- Page 1: “unlabeled”
- Page 1: “For example, in autonomous driving, the model trained using data captured in clear weather, can encounter cloudy weather followed by heavy rain, etc. during deployment.” -> remove comma before “can”
- Page 3: “Recent methodsDöbler” space missing
- Page 3: “Sppecifically,”
- Multiple occasions: “a” missing, for example: “Recent methodsDöbler et al. (2023); Brahma & Rai (2023) which have achieved better performance than CoTTA also use teacher student framework to update the adapting model.” -> “a” teacher-student framework
- Commas are missing when multiple references are cited.
The authors should use \citet and \citep instead of \cite to improve readability.

---

> ### Author Response · Authors · 2023-08-26
> **Response to Reviewer noN9 (1/2)**
>
> We thank the reviewer for the insightful and constructive comments and questions. We will now answer each of them as follows:-
>
> > It is not clear to me whether the authors used 10 corruption orderings as has been done in CoTTA. I wonder whether the evaluation might suffer from a known bug that robustbench only gives you 5k examples when loading ImageNet-C
>
> Response: The reported results are on one corruption order left to right in Table 1. We thank the reviewer for pointing out the bug in the robustbench setup. We thought it was the chosen experimental protocol by CoTTA. We present a new table below with this issue rectified (we use all 50,000 samples and average over 10 orders). We observe that the performance of the proposed framework is quite stable for different orders of the test dataset and for longer time scales.
>
> | Method | Avg Error +/- Std Dev |
> |-|-|
> |CoTTA| 64.61 +/- 2.72|
> |SANTA| 60.18 +/- 0.13|
>
> > The authors will need to run this experiment as no ETA (lifelong) results were included in the ETA paper.
>
> Response: As suggested, we ran ETA for this setting, and the results compared to the proposed SANTA framework are presented below.
> We observe that for CIFAR-10C and CIFAR-100C, our model significantly outperforms ETA. On the ImageNet-C dataset, SANTA performs similarly to ETA. We will add these results to the manuscript.
>
> |Method|CIFAR-10C|CIFAR-100C|ImageNet-C|
> |-|-|-|-|
> |ETA|18.1|32.1|60.2|
> |SANTA|16.1|30.3|60.1|
>
>
> > Some interpretations of the results are wrong: The authors write that their method outperforms all other methods, which is not true, looking at the numbers in the Table. “We observe that for all the datasets, the proposed SANTA outperforms all the other existing approaches.” -> This is not true: PETAL outperforms SANTA on CIFAR10-C, and SANTA performs within 1 p.p. better than RMT on CIFAR100-C and ImageNet-C. I don’t think this is even significant.
>
> We thank the reviewer for pointing it out. We will correct it in the revised manuscript.
>
> > Figure 3: The BN adapt numbers look strange.
>
> We used the official code used by CoTTA and later cross-checked it with the official code of RMT, and we got the same performance results as reported in our manuscript. We use the N = 0 version of BN adapt (refer to Eq 3 of Schneider et al.[1]). We looked at Figure 1 of Schneider et al., and the green dashed (- -) line is the closest to our baseline case. Using a graph digitizer, we see that batch sizes 8 and 64 correspond to 73mCE and 63mCE, respectively. However, mCE is not a percentage. It is defined in Eq 1[1] (it is the error rate divided by the error rate of AlexNet, which is a number less than 100). This can also be seen by the fact that the Y-axis has values beyond 120, so the graph scales up the difference. Our numbers for 8 and 64 are 76.7\% and 68\%, respectively, which is worse than the numbers presented in Schneider et al. because they do across severity, and we follow CoTTA and only perform evaluation on severity level 5. Hence, we believe the numbers presented in the paper are correct and comparable.
>
> |Method|8|16|32|64|
> |-|-|-|-|-|
> |BN_Schneider (mCE) +/- 1 |73| 67| 64|63|
> |BN_ours (Error rate) +/- 0.1 |77.2|71.0|69.6|68.5|
>
> To reiterate the difference between BN_Schneider and BN_ours:
>
> - BN_Schneider [1] uses mCE, BN_ours uses error rate. We expect the difference between any two values from BN_Schneider to be scaled up by 1.2 to 1.3 times on BN_ours.
> - BN_Schneider evaluates on all severity and, therefore, has a better average than BN_ours.
>
> Disclaimer: Graph digitizers can have a lot of calibration errors; therefore, we report BN_Schneider with zero significant digits after the decimal point.
>
> [1] Schneider et al. "Improving robustness against common corruptions by covariate shift adaptation." NeurIPS, 2020
>
> > Expanding on the need for Contrastive Learning
>
> Utilizing augmentations in adaptation promotes the model's resilience against variations in image space [2]. Moreover, as the model is initially trained using source data, achieving accurate classification for target data necessitates aligning its features with those learned from the source. This alignment bridges the gap between the two domains. The integration of prototypes as a perspective within contrastive learning contributes to this process. Prototypes serve as representative source exemplars, aiding in creating a cohesive feature space where similar instances are brought closer while distinct ones are pushed apart. This approach effectively enhances the model's ability to correctly classify target data by ensuring its features adhere to the patterns established during training on the source data.
>
> [2] Chen, Ting, et al. "A simple framework for contrastive learning of visual representations." ICML, 2020.

---

> > ### Author Response · Authors · 2023-08-26
> > **Response to Reviewer noN9 (2/2)**
> >
> > > Evaluation on Continuously Changing Corruptions Benchmark.
> >
> > We thank the reviewer for bringing this interesting and very practical benchmark to our attention. We are also excited to see how our method performs on it. We are currently setting up the benchmark and will get back to you once we have the results.
> >
> > > Some related works have been missed and should be included; see list below.
> >
> > We thank the reviewer for referring us to these works. We will include them in the revised manuscript.
> >
> > > Limitations:
> >
> > - Our method uses source prototypes which may not be always available, in which case, the accuracy of our method drops slightly.
> > - Single sample test-time adaptation is another practical scenario, for which our method can potentially fail. Our method relies on correcting BN statistics and contrastive learning. However, the BN statistics cannot be calculated in this case and contrastive learning cannot be uses as there would be no negatives available.
> >
> > > Minor changes
> >
> > We thank the reviewer for pointing out the typos. We will correct them in the revised manuscript. We also thank you for the suggestions to "Include citations in Table 1", "Adding ylabel to the subplot in Figure 3". We will also add the full form of "SFDA".

---

> > > ### Comment · Reviewer_noN9 · 2023-08-28
> > > **Response to the rebuttal / revision**
> > >
> > > Dear authors,
> > >
> > > thank you for responding to my concerns during the revision phase.
> > >
> > > The bug with using only 5k images really is a bug and is due to how the evaluation on ImageNet-C in Robustbench has been set up to speed up evaluation, and this is what the CoTTa authors have used out of the box. There is absolutely no reason to evaluate on 5k images instead of 50k, and I don't think the CoTTa authors wrote anything about it in their paper? It is very likely a problem because test-time adaptation to 5k images should be easier than to 50k images. The CCC benchmark shows that TTA methods fail across long adaptation time-scales. And, at the very least, it is confusing because now some papers evaluate on 5k and other papers evaluate on 50k. I would really appreciate it if you could help me bring attention to this issue such that not more papers produce wrong and inconsistent results. It would be great if you could add a footnote or a sentence in the main text stating that you needed to modify the CoTTa evaluation code to evaluate on the full 50k images because robustbench gave you 5k by default. Or at least clearly state that your evaluation uses 50k images across 10 random orderings.
> > >
> > > Thanks for all the new results. It is nice to see that your method performs better than ETA. Thank you for your commitment checking the BN adaptation numbers with a graph digitizer!
> > >
> > > I am very curious to see the numbers on CCC. I think this is the last remaining concern from my side. The issue is that it looks like that most TTA techniques fail when adapted across long time-scales.
> > >
> > > TMLR allows the authors to upload a revised version of their paper. It would be great if the authors could upload a new version with the modifications having a different color such that I could easily parse the difference. I would also like to see the revised related work section which includes the papers I suggested.
> > >
> > > Best,
> > > Reviewer noN9

---

> > > > ### Author Response · Authors · 2023-09-19
> > > > **Results on CCC benchmark**
> > > >
> > > > We thank the reviewer once again for pointing us to the CCC dataset, as we learned a lot about the present CTTA method. We are pleased to report that our method doesn’t collapse on CCC hard. It took us some time to generate the data, and it is still not done. However, we experimented with three transition speeds 1000, 2000, 5000 with one seed i.e. 43 and presented its results. The other methods are also run on the same subset of data and hence are comparable.
> > > >
> > > >
> > > > | Method  | speed-1k    | speed-2k     | speed-5k     | Mean   |
> > > > |---------|-------------|--------------|--------------|--------|
> > > > | BNStats | 6.72 +- 5.1 | 6.30 +- 5.1  | 5.93 +- 5.2  | 6.32   |
> > > > | EATA    | 8.61 +- 6.3 | 8.02 +- 6.7  | 7.79 +- 6.8  | 8.14   |
> > > > | RDumb   | 9.48 +- 9.9 | 9.66 +- 10.4 | 9.21 +- 10.6 | 9.45   |
> > > > | SANTA   | 9.80 +- 6.5 | 8.99 +- 6.5  | 8.22 +- 6.5  | 9.00   |
> > > >
> > > > TABLE: Mean and standard deviation of batchwise accuracy on subset of CCC-Hard.
> > > >
> > > > **Observations:**
> > > >
> > > > - CoTTA, TENT, ETA all collapsed as shown in the RDumb paper and therefore we didn't perform more experiments with them.
> > > >
> > > > - Interestingly but also expectedly, RDumb also collapses sometimes, but due to reset it gets back to source and is functional again. In other words, in some stretches, RDumb has batch accuracy in the range of 25-35% which is very good but then it collapses to 0% and stays there till the next reset. This results in the high standard deviation as seen above.
> > > >
> > > > - SANTA on the other hand has a lower standard deviation which is more desirable.

---

> > > > > ### Comment · Reviewer_noN9 · 2023-09-21
> > > > > **Response on the CCC results**
> > > > >
> > > > > Dear authors,
> > > > >
> > > > > thank you for taking the time and testing on the CCC benchmark. The reported results are strong and show that SANTA indeed does not collapse when exposed to changing corruptions across longer time-scales. This is very good and interesting since most other TTA methods do collapse. I would encourage you to also test on the "CCC medium" case as this should be easier and maybe more representative for the final paper version.
> > > > >
> > > > > Tiny nitpick: in research question 7, you should take a random set of noises, not the first 7 noises. And possibly repeat using different sets of k random noises.
> > > > >
> > > > > Otherwise, my concerns have been resolved and I will be recommending to accept this paper.
> > > > >
> > > > > Best,
> > > > > Reviewer noN9

---

> > > > > > ### Author Response · Authors · 2023-09-21
> > > > > > **Thank you!**
> > > > > >
> > > > > > Dear Reviewer noN9,
> > > > > >
> > > > > > We appreciate your feedback on our paper and your thoughtful insights regarding our work and introducing us to the CCC benchmark.
> > > > > >
> > > > > > We have uploaded a revised version, with all the suggested changes highlighted in blue.
> > > > > >
> > > > > > We are still generating the CCC medium data. We will certainly report the results as early as we can and incorporate it in the final revision.
> > > > > >
> > > > > > Regarding your feedback on research question 7, we understand and accept your point about selecting a random set of noises rather than the first 7. However, our objective was to perform a simple experiment to demonstrate the generalization capability of TTA frameworks. Moreover, in the chosen order, the first 7 corruptions come from two corruption families, namely noise and blur. We then evaluate it on other corruptions from the family natural and digital. This is infact a hard sequence to evaluate generalization, as we now have a model that is adapted on two families of corruption, whose performance is evaluated on unseen corruption families. Therefore, we argue that the performance in the above experiment is representative of the generalizability of our framework, even though it may not be statistically robust as you correctly pointed out.
> > > > > >
> > > > > > We value your evaluation and your recommendation to accept our paper. Your feedback has contributed to enhancing the quality of our research, and we appreciate your support.
> > > > > >
> > > > > > Thank you for your time and consideration.
> > > > > >
> > > > > > Best regards,
> > > > > >
> > > > > > Authors

---

> > > > > > > ### Comment · Reviewer_noN9 · 2023-09-21
> > > > > > > **Short response**
> > > > > > >
> > > > > > > Dear authors,
> > > > > > >
> > > > > > > considering question 7: yes, this makes sense and is completely fine. It's not critical to run the suggested experiment.
> > > > > > >
> > > > > > > It's fine if the medium dataset will not be ready until the discussion period closes. Please include them in the final version and it's all good.
> > > > > > >
> > > > > > > Best,
> > > > > > > Reviewer noN9

---

### Review · Reviewer_qTGo · 2023-08-30

**Summary Of Contributions:**

The paper introduces the challenge of adapting models to changing test environments and presents the SANTA framework. SANTA adapts online with key features like effectiveness for different batch sizes, maintaining source domain performance, and minimal parameter requirements. It achieves this by adjusting batch normalization parameters using source anchoring self-distillation. Additionally, a source-prototype contrastive alignment is proposed to group target samples naturally while retaining learned semantics. Extensive evaluations on tough benchmarks validate SANTA's applicability for real-world scenarios.

**Audience:**

Yes

**Claims And Evidence:**

Yes

**Requested Changes:**

Here are some suggested modifications: emphasize the differences between the proposed method and existing ones, enhance readability, and increase theoretical explanations or interpretability

**Strengths And Weaknesses:**

Strengths:

Important research problem: The paper focuses on a crucial problem related to continual test time adaptation. Continual learning and adaptation are essential in dynamic environments, where the distribution of data may change over time. The relevance of the problem is a major strength, as it contributes to enhancing the practical applicability of machine learning models.

Comprehensive framework: The proposed framework, involving the Source Anchoring Network and Target Alignment, demonstrates a holistic approach to addressing the challenges of continual test time adaptation. This comprehensive methodology adds value to the paper by providing a complete solution that tackles multiple aspects of the problem.

Experimental validation: The paper supports its claims with experiments on multiple datasets. The inclusion of diverse datasets enhances the generalizability of the proposed approach and demonstrates its effectiveness across various scenarios. This empirical validation adds credibility to the practical viability of the framework.

Weaknesses:

Lack of novelty: One notable weakness is the perceived lack of novelty in the proposed framework. The components, such as the Source Anchoring Network and Target Alignment, may not significantly deviate from existing approaches in the field. Providing a clearer differentiation between the proposed method and previous works, along with innovative elements, would enhance the paper's contribution. A meirt is that the entire method framework can address many challenges present in practical scenarios (e.g. small batch size, hyperparameter tuning, etc).

Insufficient theoretical analysis: The paper could benefit from a more comprehensive theoretical analysis. Explaining the underlying principles, mechanisms, and theoretical foundations of the proposed framework would contribute to a deeper understanding of its functioning. This analysis would elevate the scientific rigor of the paper and provide insights into why certain design choices were made.

Readability improvement: While the paper covers a complex topic, its readability could be improved. The organization of the content, clarity in explanations, and effective use of visual aids (if applicable) can significantly enhance the paper's accessibility to a wider audience. Ensuring that the paper is easy to understand for both experts and those less familiar with the topic is important for its impact.

---

> ### Author Response · Authors · 2023-09-19
> **Response to Reviewer qTGo (1/2)**
>
> We would like to express our gratitude to the reviewer for recognizing the practical applicability of our method, especially in scenarios where other approaches struggle with smaller batch sizes.
>
> >Lack of novelty: One notable weakness is the perceived lack of novelty in the proposed framework. The components, such as the Source Anchoring Network and Target Alignment, may not significantly deviate from existing approaches in the field. Providing a clearer differentiation between the proposed method and previous works, along with innovative elements, would enhance the paper's contribution. A merit is that the entire method framework can address many challenges present in practical scenarios (e.g. small batch size, hyperparameter tuning, etc).
>
> We agree with the reviewers that individually many of the modules used by the proposed framework is inspired by the existing seminal works. The main contributions of the approach as well as the difference with the existing approaches can be summarised as follows:
>
> - None of the existing approaches, which use one or more of the similar modules, possess all the desirable properties of the CTTA setting. Thus, integrating the right modules for this task is one of the main contributions of this work. Many of the design choices were not obvious. For example, it is not obvious that replacing the stochastic restoration (which is used by multiple CTTA frameworks) by simple source anchoring will result in an equally effective, if not better approach.
>
> - Teacher student model has been a default choice for majority of the current approaches like CoTTA and RMT, where the teacher (superior) predicts the classes, while the student learns about the current domain. This teacher student framework also requires a momentum component, which, while not only is data dependent, but also requires storage of the teacher and student model, thereby increasing the storage requirements. We are the first to replace this entire module using a single model, which can continuously be updated. Our source anchoring also ensures that our method remains robust and does not collapse under any circumstances. We believe that this stability is a significant advantage, particularly in practical applications where consistency and reliability are paramount.
>
> - Our method is very effective for lower batch sizes. This is primarily because we source anchor the model while also updating fewer parameters. While TENT also updates only BN parameters, it is still prone to catastrophic forgetting. In our case, the objective used will not let the model drift much from the original model. Hence, it mitigates overfitting, especially when updating the model using small batch sizes.
>
> - We would like to emphasize that the strength and contribution of the approach lies in its simplicity and its effectiveness. Further, we have explored the Continually Changing Corruptions (CCC) Benchmark [A], referred to us by Reviewer noN9, during this review process. This is an even more realistic continual setting, simulating lifelong TTA. Our method has proven to seamlessly work even in such scenario. Please refer to our response 'Summary of Rebuttal' for a brief overview of CCC and our results.
>
> [A] Press et al. “RDumb: A simple approach that questions our progress in continual test-time adaptation”
>
> >Readability improvement: While the paper covers a complex topic, its readability could be improved. The organization of the content, clarity in explanations, and effective use of visual aids (if applicable) can significantly enhance the paper's accessibility to a wider audience. Ensuring that the paper is easy to understand for both experts and those less familiar with the topic is important for its impact.
>
> Thank you for this suggestion. As suggested, we will go through the paper carefully and try to improve the clarity.

---

> > ### Author Response · Authors · 2023-09-19
> > **Response to Reviewer qTGo (2/2)**
> >
> > >Insufficient theoretical analysis: The paper could benefit from a more comprehensive theoretical analysis. Explaining the underlying principles, mechanisms, and theoretical foundations of the proposed framework would contribute to a deeper understanding of its functioning. This analysis would elevate the scientific rigor of the paper and provide insights into why certain design choices were made.
> >
> > In our approach, we leverage the paper [B] as a foundational piece of theoretical support for the adoption of knowledge distillation. Specifically, we employ the source weights with adaptive BNStats [C], as our teacher model and connect this choice to the insights provided by [B]. In BNStats, the authors convincingly demonstrate that using adaptive BN statistics is effective for enhancing the robustness of models against common corruptions. This observation has made it a fundamental component of numerous methods including, but not limited to TENT, CoTTA, RMT. Accordingly, we incorporate this BNStats model as the teacher within our framework, aligning with the rationale outlined in [B].
> >
> > Now, when interpreting the findings of [C], we view it as a regularization technique enabling us to identify features capable of mitigating the impact of domain shifts, thereby yielding immediate improvements in classification performance. Building on this understanding, within the context of [B], we can assert that the eigen basis responsible for encoding class-related information corresponds to the largest eigenvalues, while the eigen vectors associated with domain-related information correspond to smaller eigenvalues. Consequently, when we perform knowledge distillation from the source model to our adapting model, we are effectively emphasizing the class-related information in the feature vector while introducing some confusion between the domain aspects. This process enhances the performance of the adapting model beyond that achieved by the BNStats model. For a more mathematically rigorous treatment of the above statement refer to Theorem 5 of [B].
> >
> > Further, to support our interpretation of BNStats as a regularization, we present the following experiment. Given the features produced by a model when it is used on some dataset, we empirically show that the vector corresponding to the largest eigen value for BN Stats Adapt indeed corresponds to the class information of the corresponding features. Therefore amplifying this underlying regularization will always yield a better model. The protocol and results of the experiments are as follows:
> >
> > **Protocol:**
> >
> > 1. Get the features of CIFAR-10C dataset passed through the model (Source or BNStats).
> > 2. Compute the mean feature vectors w.r.t. the domain of the feature vectors (15 vectors corresponding to 15 corruptions). The first eigenvector of these vectors should be the direction of domain information ($v_{dom}$).
> > 3. Compute the mean feature vectors w.r.t. the class of the feature vectors (10 vectors for 10 classes). The first eigenvector of these vectors should be the direction of class information ($v_{cls}$).
> > 4. Now, for all the feature vectors collected in Step (1), find the PCA directions and compare them with the $v_{dom}$ and $v_{cls}$.
> >
> > **Result:**
> >
> > Shape of all_features = torch.Size([150000, 640])
> >
> > Shape of class_protos = torch.Size([10, 640])
> >
> > Shape of domain_protos = torch.Size([15, 640])
> >
> > **For Source features:**
> >
> > _CosSim_(first eigenvector of all_features, first eigenvector of class_protos) = 0.27756
> >
> > _CosSim_(second eigenvector of all_features, first eigenvector of class_protos) = 0.88336
> >
> > _CosSim_(first eigenvector of all_features, first eigenvector of domain_protos) = 0.95310
> >
> > _CosSim_(second eigenvector of all_features, first eigenvector of domain_protos) = 0.25005
> >
> > **For BNStats features:**
> >
> > _CosSim_(first eigenvector of all_features, first eigenvector of class_protos) = 0.98311
> >
> > _CosSim_(second eigenvector of all_features, first eigenvector of class_protos) = 0.04123
> >
> > _CosSim_(first eigenvector of all_features, first eigenvector of domain_protos) = 0.14569
> >
> > _CosSim_(second eigenvector of all_features, first eigenvector of domain_protos) = 0.01475
> >
> > **Observation:**
> >
> > We perform this analysis using the Source model and the BN Stats model.
> >
> > - For the source model, we observe that the first principal component of the data aligns with the $v_{dom}$ whereas the second component aligns with $v_{cls}$. This is as expected and therefore, the original source model cannot be used as an anchor.
> > - On the other hand, on analyzing features collected from the BNStats model, we see that the first principal component aligns with $v_{cls}$ and the other components only negligibly align with the $v_{dom}$. Therefore this is a theoretically sound candidate for being an anchor.
> >
> > [B] Mobahi et al. "Self-distillation amplifies regularization in hilbert space." NeurIPS 2020
> >
> > [C] Schneider et al. "Improving robustness against common corruptions by covariate shift adaptation." NeurIPS 2020

---

### Review · Reviewer_aE44 · 2023-09-11

**Summary Of Contributions:**

This paper tackles the problem of continuous test-time adaptation (CTTA) problem. The paper states the model should handle flexible batch size, forgetting from source and minimum tunable parameters. The authors propose to optimize batch normalization layers with source anchoring distillation loss and source-driven contrastive loss. Experiments show improvement over the comparing method and sufficient ab lation study has been done.

**Audience:**

Yes

**Claims And Evidence:**

Yes

**Requested Changes:**

In general, I think the proposed method is incremental w.r.t the existing works and also lack of justification on why it can help various batch size. I suggest the authors strength this part.

**Strengths And Weaknesses:**

Pros:
1.  The motivation and organization are clear and easy to follow.
2. The proposed method achieves on-bar or slightly better performance than the comparing methods.
3. Experiments have sufficient ablation study including the discussion of batch size, inference time/cost, different propose module and etc.

Cons:
1. Novelty
My major concern lies in the novelty of this work. First, batch-normalization optimization in CTTA has been widely explored such as TENT, [1,2]. Methods along this direction naturally have fewer parameters and better inference time.  Second, the self-distillation loss has been explored in [2]. Contrastive learning or pseudo labels with neighbor information has been discussed in [3,4]. In short, I think the paper integrates several effective techniques for relating DA topics to the CTTA task. The novelty is a bit incremental.

2. Motivation
The author states that a good TTA model 1) can work effectively for different (even small) batch sizes; 2) should continue to work well on the source domain; and 3) should have minimal tunable hyperparameters and storage requirements. As I mentioned, the batch normalization optimization method can naturally achieve (3) based on TENT, the student-teacher model can help (2) from RMT and CoTTA. Yet, the paper hasn't proposed any specific method to handle (1). The proposed method has comparable performance with CoTTA and RMT over the different sizes of batches in Figure 3. There is a lack of justification on why the proposed method can help on this feature.


[1] TTN: A DOMAIN-SHIFT AWARE BATCH NORMALIZATION IN TEST-TIME ADAPTATION, ICLR23
[2] EcoTTA: Memory-Efficient Continual Test-time Adaptation via Self-distilled Regularization, CVPR23
[3] Discovering Informative and Robust Positives for Video Domain Adaptation, ICLR23
[4] Domain Adaptation with Auxiliary Target Domain-Oriented Classifier, CVPR21

---

> ### Author Response · Authors · 2023-09-19
> **Response to Reviewer aE44 (1/2)**
>
> We thank the reviewer for appreciating the motivation of our proposed method and acknowledging the strong experimental analysis done to support it. We now address your comments.
>
> >Novelty My major concern lies in the novelty of this work. First, batch-normalization optimization in CTTA has been widely explored such as TENT, [1,2]. Methods along this direction naturally have fewer parameters and better inference time. Second, the self-distillation loss has been explored in [2]. Contrastive learning or pseudo labels with neighbor information has been discussed in [3,4]. In short, I think the paper integrates several effective techniques for relating DA topics to the CTTA task. The novelty is a bit incremental`
>
> We agree with the reviewers that individually many of the modules used by the proposed framework is inspired by the existing seminal works. The main contributions of the approach as well as the difference with the existing approaches can be summarised as follows:
>
> - TTN does explore BN parameters. However, it needs access to source data for initialization which may not feasible in CTTA setting.
> Similar to ours, EcoTTA employs the idea of self-distillation to preserve source knowledge. However, our objective proves to be better than the L1 loss used in EcoTTA, based on the empirical evidence shown in the Table below. Also, both TTN and EcoTTA have a warmup phase requiring labeled source data. It is noteworthy that our method is completely source-free. Contrastive learning helps in learning more generalized features that can aid adaptation. We perceive it as a module that can be used along with any TTA method considering its proven effectiveness. Specifically, in our case, we also use the class prototypes as a view to aid target clustering.
>
> |Method | ImageNet-C |
> |-------|------------|
> |CoTTA  | 62.6       |
> |EcoTTA | 63.4       |
> |SANTA  | 60.1       |
>
> TABLE: Error % (lower is better). Numbers taken from Table 2 of EcoTTA.
>
>
>
>
> - None of the existing approaches, which use one or more of the similar modules, possess all the desirable properties of the CTTA setting. Thus, integrating the right modules for this task is one of the main contributions of this work. Many of the design choices were not obvious. For example, it is not obvious that replacing the stochastic restoration (which is used by multiple CTTA frameworks) by simple source anchoring will result in an equally effective, if not better approach.
> - Teacher student model has been a default choice for majority of the current approaches like CoTTA and RMT, where the teacher (superior) predicts the classes, while the student learns about the current domain. This teacher student framework also requires a momentum component, which, while not only is data dependent, but also requires storage of the teacher and student model, thereby increasing the storage requirements. We are the first to replace this entire module using a single model, which can continuously be updated. Our source anchoring also ensures that our method remains robust and does not collapse under any circumstances. We believe that this stability is a significant advantage, particularly in practical applications where consistency and reliability are paramount.
>
> We would like to emphasize that the strength and contribution of the approach lies in its simplicity and its effectiveness. Further, we have explored the Continually Changing Corruptions (CCC) Benchmark [A], referred to us by Reviewer noN9, during this review process. This is an even more realistic continual setting, simulating lifelong TTA. Our method has proven to seamlessly work even in such scenario. Please refer to our response 'Summary of Rebuttal' for a brief overview of CCC and our results.
>
> [A] Press et al. “RDumb: A simple approach that questions our progress in continual test-time adaptation”

---

> > ### Author Response · Authors · 2023-09-19
> > **Response to Reviewer aE44 (2/2)**
> >
> > >Motivation The author states that a good TTA model 1) can work effectively for different (even small) batch sizes; 2) should continue to work well on the source domain; and 3) should have minimal tunable hyperparameters and storage requirements. As I mentioned, the batch normalization optimization method can naturally achieve (3) based on TENT, the student-teacher model can help (2) from RMT and CoTTA. Yet, the paper hasn't proposed any specific method to handle (1). The proposed method has comparable performance with CoTTA and RMT over the different sizes of batches in Figure 3. There is a lack of justification on why the proposed method can help on this feature.
> >
> > We agree that the proposed approach also updates the BN parameters like TENT and hence has minimal hyperparameters and storage requirements. However, there are significant differences from CoTTA and RMT, such as:
> > - Ours does not have the teacher-student model, but has a single model which can be continuously updated. It also does not require stochastic restoration. Thus this effective design choice simultaneously reduces the storage requirements and also the number of tunable hyperparameters which is an often ignored, but extremely important objective for CTTA setting.
> > - The framework seamlessly works for small batch sizes. We clarify that SANTA significantly outperforms CoTTA and RMT consistently for lower batch sizes (<=50) across datasets. The following are the results (also visually shown in Figure 3 in paper) for varying batch sizes:
> >
> >
> >
> > |            |           | C-10C |           |           | C-100C|            |           | IN-C |           |
> > |------------|-----------|-----------|-----------|-----------|-----------|------------|-----------|------------|-----------|
> > | Batch Size | 10        | 25        | 50        |10        | 25        | 50         | 8         | 16         | 32        |
> > | BNAdapt    |     25.05 |     22.04 |     21.16 |      42.1 |     37.82 |      36.36 | **76.76** |      71.02 |     69.62 |
> > | TENT       |      79.1 |     37.65 |     33.29 |     97.37 |     93.69 |      87.41 |      95.7 |      80.96 |     64.39 |
> > | CoTTA      |     51.26 |     25.41 |     18.76 |     57.04 |     40.84 |      35.00 |     96.85 |      83.26 |     65.29 |
> > | RMT        |     73.57 |     41.34 |     31.63 |     75.32 |     49.52 |      38.85 |     99.69 |      84.84 |     68.34 |
> > | SANTA      | **22.03** | **18.28** | **17.17** | **39.12** | **33.88** |  **31.94** |      78.5 |  **68.46** | **63.25** |
> >
> > TABLE: Mean error percentage (lower is better) obtained by each method at smaller batch sizes for CIFAR-10C (C-10C), CIFAR-100C (C-100C) and ImageNet-C (IN-C) datasets.
> >
> > - The primary reason which we believe makes our method effective for lower batch sizes is because of source anchoring the model while also updating fewer parameters. While TENT also updates only BN parameters, it is still prone to catastrophic forgetting. In our case, the objective used will not let the model drift much from the original model. Hence, it mitigates overfitting, especially when updating the model using small batch sizes.

---

### Author Response · Authors · 2023-09-19
**Summary of Rebuttal**

We thank all the reviewers for their insightful comments. This is a summary of the additional experiments done. We will amend this comment through the review process.

>Experiments

We have performed the following experiments during the review process.

- We use 50,000 samples (correcting the bug in Robustbench setup) and perform experiments on 10 corruption orders on ImageNet-C.
- Include ETA as baseline.
- Theoretical justification of our choice of BNStats model as the anchor.
- Experiments on the challenging CCC benchmark.


>CCC benchmark

We are very thankful for Reviewer noN9 for introducing us to the CCC benchmark.
We briefly describe the benchmark below.

Continuously Changing Corruptions (CCC) [A] is a recent benchmark proposed to
measure asymptotic performance of TTA techniques. Specifically, is a dataset for evaluating models that can adapt to changing image corruptions. It is based on the ImageNet-C dataset, which contains common image corruptions such as noise, blur, rotation, etc. However, CCC extends these corruptions by applying two corruptions at the same time with finer-grained severities to allow for smooth transitions between corruptions. For example, a noise-corruption pair could be a noisy image with some parts of the image replaced by random pixels. A blur-corruption pair could be a blurry image with some parts of the image replaced by random colors.   The default test sequence in CCC is 10X longer than ImageNet-C CTTA sequence. In this setting, [A] show that most state-of-the-art methods collapse and perform worse than a non-adapting model, including models specifically proposed to be robust to performance collapse. Even in this challenging setting, we observe that our method does not collapse with time. Please refer to the response 'Results on CCC benchmark' for more details.

[A] Press et al. “RDumb: A simple approach that questions our progress in continual test-time adaptation”

---

### Decision · Action_Editor_CAEs · 2023-10-26

**Recommendation:** Accept as is

**Comment:**

This paper proposes a new method for tackling an important research problem: continual test-time adaptation. Their method is efficient, simpler than previous approaches, entirely source-free and works well (equally well or better) than a wide range of previous baselines on various challenging datasets. It is also effective for smaller batch sizes which is important for online adaptation. The authors conduct a thorough empirical study to support their claims. The reviewers also pointed out that the paper is well-written and well-organized.

I am not concerned about reviewer comments on novelty, since novelty is not a criterion for TMLR. In addition, I agree with the authors' response to those reviewers that assembling existing components in a new way is not obvious, and neither of the existing components alone possessed all desirable components that the proposed approach does.

**Audience:**

This paper tackles the topic of continual test-time adaptation which is an important research problem of interest to several members in TMLR's audience.

**Claims And Evidence:**

This paper proposes SANTA, a method for Continual Test-time Adaptation that is entirely source-free, is efficient and effective (working equally well or better than previous approaches on a variety of challenging benchmarks) while working well even smaller batch sizes too, lending itself well to the online setting.

As observed by all reviewers, the authors have performed a very extensive experimentation and ablation studies that convincingly supports their claims. The authors also worked hard during the rebuttal, comparing against an additional baseline (ETA) and on an additional challenging benchmark (that tests the stability of TTA methods over longer time scales), as requested by reviewer noN9. The reviewer was particularly satisfied with the results and their analysis that shows that their method remains stable when several other known baselines don't.